# Effects of Screen Viewing Time on Sleep Duration and Bedtime in Children Aged 1 and 3 Years: Japan Environment and Children’s Study

**DOI:** 10.3390/ijerph19073914

**Published:** 2022-03-25

**Authors:** Takafumi Nishioka, Hideki Hasunuma, Masumi Okuda, Naoko Taniguchi, Tetsuro Fujino, Hideki Shimomura, Yasuhiko Tanaka, Masayuki Shima, Yasuhiro Takeshima

**Affiliations:** 1Department of Pediatrics, Hyogo College of Medicine, Nishinomiya 663-8501, Japan; t95541026@yahoo.co.jp (T.N.); pulmeria.n.moana@gmail.com (N.T.); the-jetset@hotmail.co.jp (T.F.); simo-ped@hyo-med.ac.jp (H.S.); qfwfq555@gmail.com (Y.T.); ytake@hyo-med.ac.jp (Y.T.); 2Department of Public Health, Hyogo College of Medicine, Nishinomiya 663-8501, Japan; hi-hasunuma@hyo-med.ac.jp (H.H.); shima-m@hyo-med.ac.jp (M.S.); 3Hyogo Regional Center for the Japan Environment and Children’s Study, Hyogo College of Medicine, Nishinomiya 663-8501, Japan

**Keywords:** sleep duration, bedtime, television, digital versatile disc player, portable electronic device

## Abstract

This study aimed to clarify the effects of television/digital versatile disc (TV/DVD) viewing time and portable electronic device (PED) usage time on sleep duration and bedtime and the difference between the effects of TV/DVD and PED on sleep. The effect of TV/DVD viewing time or PED usage time on sleep duration and bedtime was analyzed using a multiple logistic regression analysis adjusted for covariates. A total of 74,525 participants were included in the analysis, using data from Japan Environment and Children’s Study. TV/DVD viewing was not associated with short sleep duration, but PED usage was associated with short sleep duration. In addition, the risk of short sleep duration increased as PED usage time increased. We also investigated the effects of sleep habits at age 1 year on sleep at age 3 years. This study showed that late bedtime at age 1 year posed a significant risk of late bedtime at age 3 years. In summary, particular caution should be paid to PED use from a child’s health perspective, and sleep habits should be focused on bedtime from the age of 1 year.

## 1. Introduction

Sleep is important for lifelong physical, mental, and emotional health maintenance. Securing sleep duration is essential for the development of the brain and body, especially during early childhood [1,2,3]. During deep sleep, growth hormones are actively secreted, promoting neural network formation in the brain and body during childhood [4]. Melatonin secretion also begins, the amount of which rapidly increases until approximately 1 year of age, forming a day and night rhythm [5]. A reduction in sleep time during infancy and childhood causes various adverse effects, such as developmental and cognitive impairment, memory deterioration, and obesity [4,6,7,8,9,10,11,12].

The association between sleep and smartphones, portable electronic devices (PEDs), and television usage has been previously reported [13,14,15,16,17,18,19,20,21,22]. A meta-analysis of the association between screen use and health indicators in infants, toddlers, and preschoolers suggested that excessive screen time was associated with being overweight/obese and shorter sleep time among toddlers and preschoolers [23].

Screen time may affect infant sleep. There are three possible reasons why screen time can possibly affect sleep, including the following: (1) long-term use of TV/digital versatile disc (TV/DVD), smartphones, and PEDs before going to bed delays bedtime; (2) for children, stimulating programs and content stimulates the brain, and the stress hormone cortisol continues to be secreted not only during the day but also at night, making it difficult for children to fall asleep [24]; and (3) the strong light of the screen at night suppresses the secretion of melatonin, and because of circadian rhythm interruption, results in late bedtime and short sleep duration. Circadian rhythm refers to the sleep/wake rhythm, and sleep hormones, such as melatonin, are deeply involved in this rhythm. Melatonin is easily affected by light, and its secretion is suppressed by the influence of light during the day and increases at night. Suppression of melatonin secretion causes difficulty falling and staying asleep [25,26]. However, in very young children, the effect of screen light exposure on circadian rhythms is not well known.

The use of PEDs is increasing due to the global spread of the internet [27]. According to the 2019 Communications Usage Trend Survey by the Ministry of Internal Affairs and Communications of Japan, the internet usage rate increased from 46.3% in 2001 to 89.8% in 2019. The ownership rates of smartphones and PEDs were 9.7% and 7.2% and 83.4% and 37.4% in 2010 and 2019, respectively [28]. In the United Kingdom, the percentage of families owning a touch-screen device increased rapidly from 7% in 2011 to 71% in 2014 [29]. Although many studies on TV viewing and sleep have been reported, few large cohort studies investigating the effects of smartphones and PEDs on sleep have been conducted. Thus, one of the purposes of this study was to clarify the effects of TV/DVD and PEDs on sleep separately. PEDs present constant access to various media of interest to children. In addition, children can not only watch but also touch the screen and talk, making it a significantly attractive tool for children, similar to a toy. As a result, it is believed that PEDs are used for a longer time and more intensively than TV/DVD. In addition, children play with PEDs alone; thus, parents may depend on PEDs as a tool of playing children. With PED dependence, parents may reduce the time spent communicating closely with their children. PEDs can be carried, and children can secretly use them for a long period of time. For these reasons, PED usage may result in shorter sleep duration and later bedtime than TV/DVD use. The increase in household PEDs has resulted in increased PED use by children and toddlers. Furthermore, parents often use not only TV or DVD players but also PEDs to let their children play. The effects of PED usage on infants’ sleep and cognitive function is not well known. Thus, caution should be exercised when using PEDs from early infancy. Parents must understand the effects of PEDs on sleep and manage their usage in an appropriate manner, different from TV/DVD, so that children do not use PEDs for a long time [30]. This study is also important for parents in considering how to manage children’s TV/DVD and PED use.

In this study, we examined the effects of TV/DVD viewing time and PED usage time on sleep time and bedtime and the difference in the effects of TV/DVD and PEDs on sleep, using data from Japan Environment and Children’s Study (JECS). The JECS is a large-scale, nationwide, multicenter, prospective birth cohort study that commenced in January 2011. The JECS has been implemented as a national project funded by the Ministry of Environment, Japan. Its main objective is to elucidate the environmental factors affecting children’s health and development.

Several longitudinal studies examining the effect of screen time on sleep habits have been reported so far, emphasizing the long-term association between screen time and sleep habits [31,32]. Ensuring sleep time in infancy is essential for brain and body development, and long-term short sleep duration and late bedtime from infancy are serious problems in terms of growth and development. To our knowledge, there is no large longitudinal cohort study examining the effects of screen time and sleep habits from infancy; the association between screen time and sleep habits is unclear. Our study examined whether screen time and sleep habits at age 1 year affect sleep at age 3 years. This was an unprecedented study examining the long-term effects of infant screen time and sleep habits.

## 2. Materials and Methods

### 2.1. Study Design and Participants

This study was conducted using data from the JECS, which has 15 regional centers in Japan [33]. The JECS aims to elucidate the effects of environmental factors on children’s health from pregnancy to childhood. Briefly, pregnant women who visited a cooperating healthcare provider in the study area or the reception counter of a municipality office to obtain a maternal and child health handbook in their first trimester were recruited in this cohort study between January 2011 and March 2014. After registration, the data of pregnant women and children from registered pregnant women were collected using self-administered questionnaires [34]. For the children, data were collected every 6 months, from 6 months to 3 years of age, after delivery using self-administered questionnaires and data at 1, 1-1/2, and 3 years of age. The data for this study were obtained from a dataset released in October 2019 (dataset jecs-ta-20190930).

### 2.2. Questionnaires

The information on sleep duration, bedtime, and TV/DVD viewing time was obtained from the questionnaire when the children were 1 year old. Information on sleep duration, bedtime, TV/DVD viewing time, and usage time of PEDs was obtained from the questionnaire when the children were 3 years old. Sleep conditions, such as sleeping posture and place, were evaluated 1 week before the questionnaire. Sleep duration data were obtained the day before the questionnaire. In this study, sleep duration, bedtime, TV/DVD viewing time, and usage time of PEDs were used in the analyses. The questionnaire on sleep duration had boxes for 30 min intervals, from 00:00 to 24:00 the next day, and a horizontal line was drawn to indicate the time during which the child slept. The questionnaire for TV/DVD viewing time provided answer options as follows: no viewing, less than 1 h, 1 h or more and less than 2 h, 2 h or more and less than 4 h, and 4 h or more. The questionnaire for PED use provided the following options: not used, less than 1 h, 1 h or more and less than 2 h, 2 h or more and less than 4 h, and 4 h or more. The options for primary sleeping positions at 1 and 3 years of age were as follows: in the parent’s bed, in a separate bed in the parent’s room, in a bed in a separate room, and others; back, side, stomach, and no particular position; and whether the parent(s) put their child to sleep at a fixed time—yes, no. In addition, the mother’s and father/partner’s highest level of education were selected as follows: junior high school, high school, technical junior college, technical/vocational college, associate degree, bachelor’s degree, and graduate degree (master’s/doctorate). Cohabitants at 1-1/2 years old were selected as follows: grandparent(s), older siblings, younger siblings, and other(s).

### 2.3. Statistical Analyses

All statistical analyses were performed using SAS version 9.4 (SAS Institute Inc., Cary, NC, USA). The associations among TV/DVD viewing time, sleep duration, and bedtime at 1 and 3 years of age were analyzed using multiple logistic regression analysis. For viewing time, we calculated the odds ratios (ORs) of “less than 1 h”, “1 h or more and less than 2 h”, “2 h or more and less than 4 h”, and “4 h or more” compared with “no viewing” and adjusted for covariates. The association between PED usage time and sleep duration and bedtime at the age of 3 years was analyzed using multiple logistic regression analysis. For the usage time of PEDs, we calculated the OR of “less than 1 h”, “1 h or more and less than 2 h”, “2 h or more and less than 4 h”, and “4 h or more” compared with “not used” and adjusted for covariates.

The cut-off values for sleep duration were set to less than 11 and 10 h for 1 and 3 years of age, respectively. The National Sleep Foundation recommends 11–14 h for children aged 1 year and 10–13 h for children aged 3 years [35]. The bedtime cut-off value was set after 22:00, based on the data of Mindell et al. [36] and the Japanese Society of Child Health [37]. Bedtime was taken as the time the horizontal line was drawn from between 17:00 and 23:30 on the questionnaire and was regarded as the time that the child went to sleep. The covariates were the parents’ highest level of education, co-residents, sleeping place, sleeping posture, and whether they slept at a fixed time.

### 2.4. Ethics

The JECS protocol was reviewed and approved by the Ministry of the Environment’s Institutional Review Board (IRB) on Epidemiological Studies (IRB number: 100910001) and by the ethics committees of all participating institutions. Written informed consent was obtained from all the participants’ parents.

## 3. Results

A total of 104,062 fetuses were registered in the Japan Environment and Children’s Study. Of the 100,304 live births, 22,658 were excluded because of incomplete responses to the questionnaire. Of the 77,646 participants, 3121 were excluded because the data on sleep time and bedtime were unknown. Finally, 74,525 participants were included in the analyses (Figure 1).

The characteristics of the subjects are shown in Table 1. The rate of bedtime after 22:00 was as high as 16.4% at the age of 1 and further increased to 29.6% at the age of 3. This result suggests that children sleep less in Japan than worldwide, and this finding is consistent with that of a previous study [36]. Moreover, 89.5% and 98.3% of children at the age of 1 and 3 years watched TV/DVD, respectively. This confirms that TV or DVD is a significantly familiar device. Smartphones and PEDs were used by 49% of children aged 3 years. This suggests that PED is widely used not only by adults but also by children (Table 1).

### 3.1. Association between Screen Viewing Time and Average Sleep Duration and Bedtime

We classified TV/DVD and PED usage time into four groups and examined average sleep duration and average bedtime in each group (Table 2). Table 2 showed the average sleep duration of each group classified by TV/DVD and PED usage time. There was no difference in sleep duration between TV/DVD viewing and PED usage. Table 2 showed the average bedtime of each group classified by TV/DVD and PED usage time. Comparing TV/DVD viewing and PED usage, children with PED usage clearly were prone to a late bedtime. This indicates that PEDs have a stronger effect on bedtime than TV/DVD.

Next, we set cutoff values for sleep duration and bedtime and analyzed using multiple logistic regression analysis to examine whether the use of TV/DVD and PEDs poses a risk of short sleep duration and late bedtime in children (Figure 2 and Figure 3). In addition, we conducted a longitudinal survey to examine whether viewing TV/DVD at the age of 1 poses a risk of short sleep duration and late bedtime at the age of 3. Viewing TV/DVD at the age of 1 and short sleep duration and late bedtime at age 1 were examined for their effect on sleep at age 3 (Figure 4 and Figure 5).

### 3.2. Association between Screen Viewing Time and Short Sleep Duration 

Figure 2 presents the OR of short sleep duration compared to children with no TV/DVD viewing and no PED usage. TV/DVD viewing was not associated with short sleep duration at age 1 and 3 years. With PED, the OR increased as the usage time increased. This result showed that PED usage has a stronger effect on sleep than TV/DVD viewing, similar to the result in Table 2.

### 3.3. Association between Screen Viewing Time and Late Bedtime 

Figure 3 presents the OR of late bedtime compared to children with no TV/DVD viewing and no PED usage. For TV/DVD, the longer the viewing time, the higher the OR for both 1 and 3 years old. Similarly, for PED usage, the OR increased with usage time. These results suggest that the longer the TV/DVD and PED are used, the higher the risk of late bedtime.

### 3.4. Effects of Screen Viewing Time and Sleep Habits at Age 1 Year on Short Sleep Duration at Age 3 Years 

Figure 4 presents whether TV/DVD viewing, short sleep duration, and late bedtime at age 1 year affect short sleep duration at age 3 years. We presented the ORs for short sleep duration at age 3 years. There was no association between TV/DVD viewing at age 1 year and short sleep duration at age 3 years. Children with short sleep duration at the age of 1 year had high OR (2.21, (95% confidence interval (CI), 2.04–2.39)). Children with late bedtime at the age of 1 year had a high OR (1.20 (95% CI, 1.11–1.29)). This result indicates that securing sleep time without staying up late from early infancy leads to the establishment of healthy sleep habits in the long term.

### 3.5. Effects of Screen Viewing Time and Sleep Habits at Age 1 Year on Late Bedtime at Age 3 Years

Figure 5 presents whether TV/DVD viewing, short sleep duration, and late bedtime at age 1 year affect late bedtime at age 3 years. We presented the ORs for late bedtime at age 3 years. TV/DVD viewing at the age of 1 year had higher ORs as the viewing time increased. This suggests that TV/DVD viewing at age 1 year is associated with late bedtime at age 3 years. Children with late bedtime at the age of 1 year had a significantly high OR for late bedtime at the age of 3 years (3.60 (95% CI, 3.46–3.75)). This suggests that late bedtime at age 1 year poses a significant risk of late bedtime at the age of 3 years.

## 4. Discussion

For the average bedtime at the age of 1 and 3 years, there was approximately a 20 min later difference in the group watching TV/DVD for 4 h or more than in the group not watching TV/DVDs. The same applied to PEDs; there was a difference of approximately 20 min later in the group using PEDs for 2 to 4 h or more compared with the group not using PEDs. The TV/DVD viewing time and PED usage time had an influence on bedtime. The difference in bedtime between the group not using PEDs and the group with 4 h or more was small, approximately 10 min. This may be because the group with a PED usage time of 4 h or more comprised only 0.3% of the total population.

This study showed that the sleep habits at the age of 1 year were associated with the children’s sleep habits when they were 3 years old. Thus, it is necessary to pay attention to screen use and sleep habits (sleep duration and bedtime) from the age of 1 year. In this study, we examined the effects of screen time on sleep in children up to the age of 3 years, but it is necessary to follow up with a longitudinal survey in the future.

A study reported that 6-month-old infants exposed to screen media had lower sleep duration compared to unexposed infants [38]. For children aged 6 months to 3 years, excessive tablet usage has been reported to affect sleep duration and cause sleep delay. In addition, TV exposure did not reduce sleep time and delay bedtime [39]. A meta-analysis and systematic review that investigated the association between screen use and health indicators for infants, toddlers, and preschoolers showed a strong association between excessive screen use and a shortening of sleep duration [23]. According to a review, 90% of previous studies related to sleep and screen media use suggested that screen media use is associated with delayed sleep duration and decreased sleep duration in children [40]. A study of infants aged 0–18 months showed that infants exposed to a touchscreen during the day had less nap time with increased nighttime sleep, but exposure to TV had no effect on sleep during the day and at night compared to children exposed to touchscreens [41]. In this study, the association between screen viewing time and sleep duration and bedtime was clarified in children. Currently, the effect of screen light exposure on circadian rhythms is not well known in very young children. Circadian rhythm is composed of a complex neural network centered on the suprachiasmatic nucleus of the hypothalamus, and various hormones and neurotransmitters are involved. Various studies have been conducted to elucidate the mechanism of circadian rhythm, and recently the involvement of melanopsin and dopamine has been reported [42,43]. Further studies are needed to examine the effects of light exposure on circadian rhythms in very young children.

Comparing TV/DVD viewing and PED usage in this study, children with PED usage were associated with late bedtime, and this suggests that PEDs have a stronger effect on bedtime than TV/DVD. In addition, approximately half of the participants in this study used PEDs. This means that PEDs are usually used for children; if PEDs cause adverse effects on sleep, it may result in serious problems in maintaining children’s health. 

Similar to our study, previous studies have compared the use of PEDs with TV viewing on sleep and reported that PED use caused a greater reduction in sleep duration and delayed bedtime than TV viewing [39,41]. One of the reasons for this is possibly the degree of dependence, not the use time. Regarding TV and DVD viewing, it is possible that the children did not focus on what they were watching. However, PED use can require more attention if the child is playing a game or using a program, which may hold a child’s interest. Therefore, there is a possibility that children using PEDs keep watching the screen enthusiastically, even for a short time, possibly resulting in a higher dependence. Even in the clinical setting of pediatrics, we often observe a hospitalized child playing alone by operating a PED. In addition, the COVID-19 pandemic has significantly changed our lifestyles. The number of papers assessing the association between COVID-19 and social media addiction is increasing [44,45,46,47,48,49], and the connection between humans and the internet will become even stronger in the future.

This study has a limitation in that since the evolution of and substantive uptake of web-enabled PED devices such as smartphones among children is relatively recent, the data collected before 2016 may be under-reporting current trends [50,51]. At the age of 3 years, PED usage was associated with short sleep duration and late bedtime. However, causality cannot be assumed from the present study. The issue of causality is raised as a limitation because some children may have a naturally delayed circadian rhythm. Another limitation of this study was that the sleep duration was evaluated using a questionnaire completed by the caregivers, and the data for sleep time was only from the day before completing the questionnaire. Nap time was not considered in this study because it is difficult to distinguish between nap and night sleep in 1-year-old children. We also did not investigate whether the children were always at home or in a nursery, or whether there was a TV in the children’s rooms. Children who stay in daycare centers tend to wake up early and have less sleep duration than children who remain at home, regardless of electronic media usage [52]. A report showed that there is an association between the presence of TV/electronic media in children’s rooms and sleep quality [18].

## 5. Conclusions

TV/DVD viewing was not associated with short sleep duration, but PED usage was associated with short sleep duration. In addition, the risk of short sleep duration increased as PED usage time increased. This result indicates that caution should be paid in PED use from the viewpoint of pediatric health. Few large cohort studies have previously examined the association between screen viewing time and sleep in children, and the results of this study are important. We also examined the effects of sleep habits at age 1 year on sleep at age 3 years. This study showed that late bedtime at age 1 year posed a significant risk of late bedtime at age 3 years. These results indicate that attention should be paid to bedtime to acquire healthy sleep habits from infancy.

## Figures and Tables

**Figure 1 ijerph-19-03914-f001:**
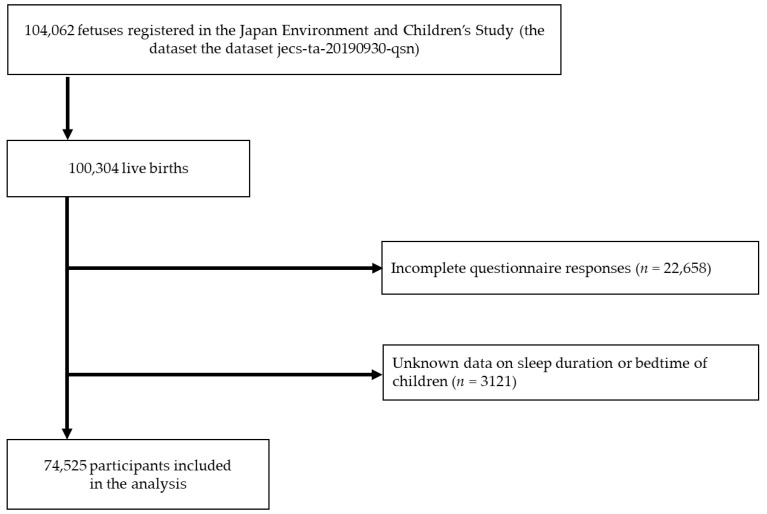
Flowchart of participants.

**Figure 2 ijerph-19-03914-f002:**
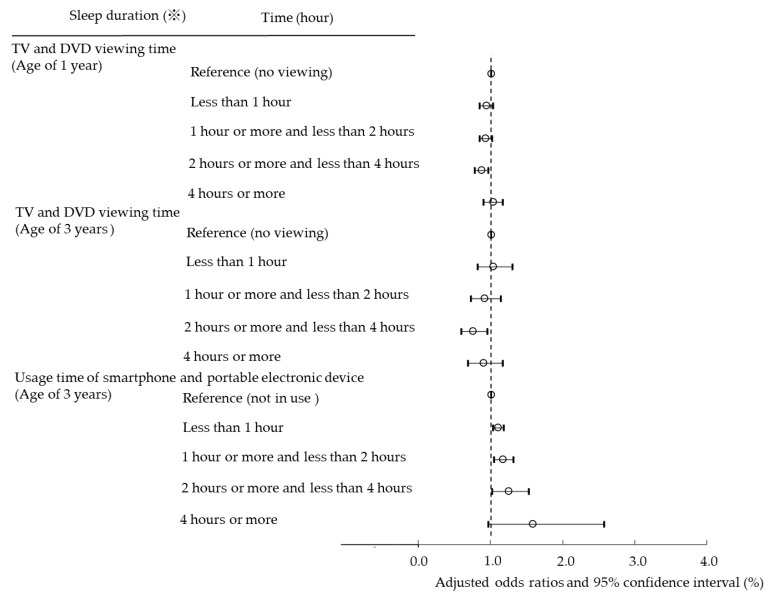
Association among TV/DVD viewing time at the age of 1 and 3 years, PED usage time at the age of 3 years, and sleep duration by multiple logistic regression analysis.※ Sleep duration at the age of 1 year is less than 11 h. Sleep duration at the age of 3 years is less than 10 h.

**Figure 3 ijerph-19-03914-f003:**
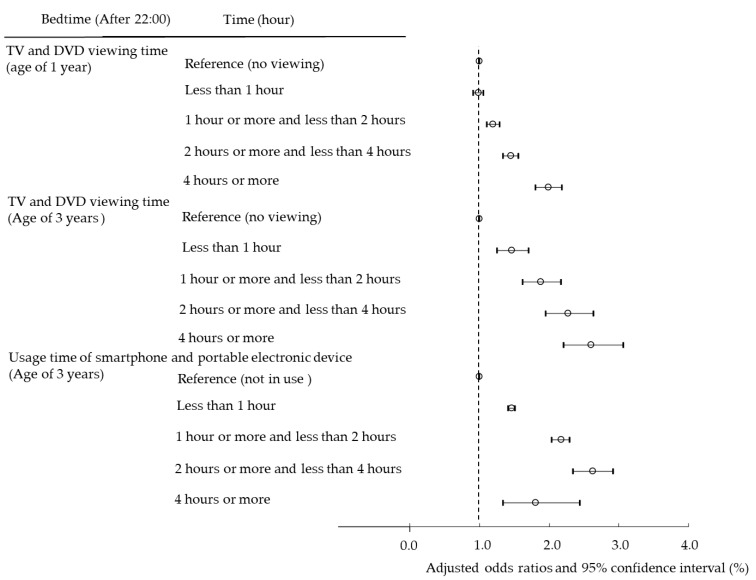
Association among TV/DVD viewing time at the age of 1 and 3 years, PED usage time at the age of 3 years, and bedtime by multiple logistic regression analysis.

**Figure 4 ijerph-19-03914-f004:**
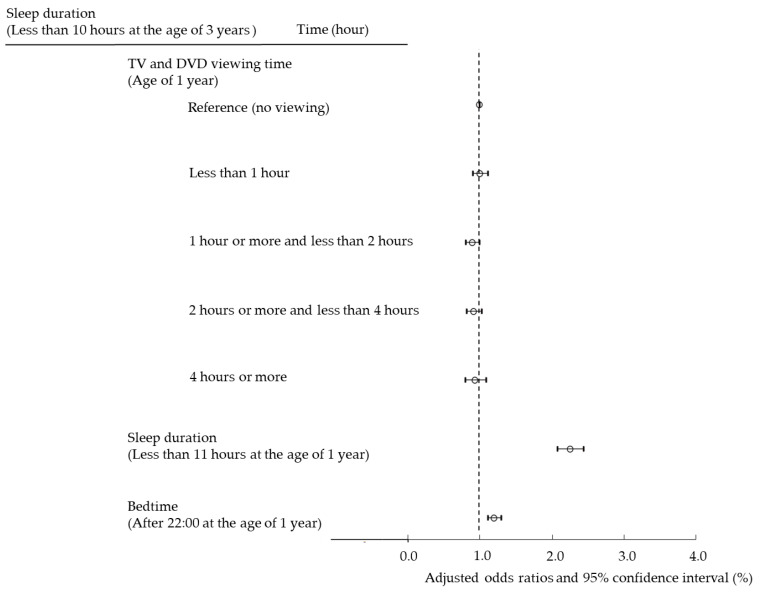
Effect of TV/DVD viewing time, sleep duration, and bedtime at the age of 1 year on sleep duration (less than 10 h) at the age of 3 years.

**Figure 5 ijerph-19-03914-f005:**
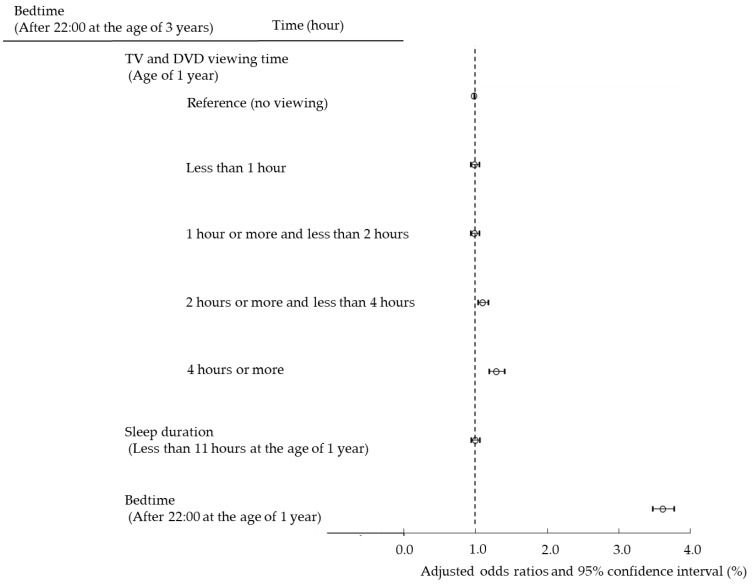
Effect of TV/DVD viewing time, sleep duration, and bedtime at the age of 1 year on bedtime (after 22:00) at the age of 3 years.

**Table 1 ijerph-19-03914-t001:** Characteristics of the participants (*n* = 74,525).

	Age of 1 Year	Age of 3 Years
Sleep duration		
11 h or more	67,970 (91.2%)	
Less than 11 h	6555 (8.8%)	
Sleep duration		
10 h or more		69,447(95.8%)
Less than 10 h		5078 (4.2%)
Bedtime		
Before 22:00	62,316 (83.6%)	52,457 (70.4%)
After 22:00	12,209 (16.4%)	22,068 (29.6%)
TV and DVD viewing time		
No viewing	7817 (10.5%)	1278 (1.7%)
Less than 1 h	25,190 (33.8%)	18,056 (24.2%)
1 h or more and less than 2 h	22,268 (29.9%)	33,230 (44.6%)
2 h or more and less than 4 h	14,559 (19.5%)	18,699 (25.1%)
4 h or more	4691 (6.3%)	3262 (4.4%)
Usage time of smartphone and portable electronic device		
Not in use		38,033 (51.0%)
Less than 1 h		29,159 (39.1%)
1 h or more and less than 2 h		5659 (7.6%)
2 h or more and less than 4 h		1482 (2.0%)
4 h or more		192 (0.3%)
Sleeping place		
Beds and futons with parents	56,610 (76.0%)	62,544 (83.9%)
Another bed, futon in the room where parents sleep	17,404 (23.4%)	10,987 (14.7%)
Bed and futon in another room from parents	417 (0.6%)	731(1.0%)
Other	94 (0.1%)	263 (0.4%)
Sleeping posture		
Supine	22,159 (29.7%)	27,902 (37.4%)
Sideways	24,752(33.2%)	19,696 (26.4%)
Prone	15,978 (21.4%)	9956 (13.4%)
Not determined	11,636 (15.6%)	16,971 (22.8%)
Sleep at a fixed time		
Yes	69,660 (93.5%)	70,026 (94.0%)
No	4865 (6.5%)	4499 (6.0%)
Family living together other than parent		
Grandparents		15,428 (20.7%)
Older sibling		39,736 (53.3%)
Younger sibling		2841 (3.8%)
Other		5812 (7.8%)
Mother’s educational background		
Junior high school		2617 (3.5%)
High school		22,082 (29.6%)
Technical junior college		1212 (1.6%)
Technical/Vocational college		17,316 (23.2%)
Associate degree		13,722 (18.4%)
Bachelor’s degree		16,386 (22.0%)
Graduate degree (master’s/doctorate)		1190 (1.6%)

**Table 2 ijerph-19-03914-t002:** The average sleep duration and bedtime according to TV/DVD viewing time and PED usage time.

	**Sleep Duration (min)**					
	**Age of 1 Year**			**Age of 3 Years**		
	** *n* **	**mean**	**SD**	** *n* **	**mean**	**SD**
TV and DVD viewing time						
No viewing	7817	777.6	92.4	1278	700.8	75.0
Less than 1 h	25,190	779.4	90.6	18,056	700.2	72.6
1 h or more and less than 2 h	22,268	777.6	89.4	33,230	697.8	70.8
2 h or more and less than 4 h	14,559	777.0	89.4	18,699	697.2	69.6
4 h or more	4691	772.8	93.0	3262	693.0	72.0
Usage time of smartphone and portable electronic device						
Not in use				38,033	699.6	70.2
Less than 1 h				29,159	697.2	71.4
1 h or more and less than 2 h				5659	694.8	74.4
2 h or more and less than 4 h				1482	692.4	75.6
4 h or more				192	693.0	88.2
		**Bedtime (h:min)**				
	**Age of 1 Year**			**Age of 3 Years**		
	** *n* **	**mean**	**SD**	** *n* **	**mean**	**SD**
TV and DVD viewing time						
No viewing	7817	20:48	00:54	1278	21:06	00:46
Less than 1 h	25,190	20:50	00:54	18,056	21:17	00:44
1 h or more and less than 2 h	22,268	20:53	00:56	33,230	21:21	00:47
2 h or more and less than 4 h	14,559	20:57	00:56	18,699	21:24	00:52
4 h or more	4691	21:04	01:07	3262	21:29	00:59
Usage time of smartphone and portable electronic device						
Not in use				38,033	21:15	00:46
Less than 1 h				29,159	21:24	00:48
1 h or more and less than 2 h				5659	21:33	00:53
2 h or more and less than 4 h				1482	21:38	01:02
4 h or more				192	21:27	01:14

## Data Availability

Data are unsuitable for public deposition due to ethical restrictions and the legal framework of Japan. It is prohibited by the Act on the Protection of Personal Information (Act No. 57 of 30 May 2003, amendment on 9 September 2015) to publicly deposit data containing personal information. Ethical Guidelines for Medical and Health Research Involving Human Subjects enforced by the Japan Ministry of Education, Culture, Sports, Science and Technology and the Ministry of Health, Labour and Welfare also restrict the open sharing of the epidemiological data. All inquiries about access to data should be sent to: jecs-en@nies.go.jp. The person responsible for handling inquiries sent to this e-mail address is Shoji F. Nakayama, JECS Programme Office, National Institute for Environmental Studies.

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
