# Peer review of "Effects of Screen Viewing Time on Sleep Duration and Bedtime in Children Aged 1 and 3 Years: Japan Environment and Children’s Study"

_ijerph, 2022, doi:10.3390/ijerph19073914_

Round 1

Reviewer 1 Report

Introduction: The authors have undertaken an analysis of third party data examining 2011-14 PED use and sleep in 1-3 year olds. This is an important study with important health implications. The authors may wish to consider the following points:

Introduction

  • Please expand the introduction and discuss the possible reasons why screen time may impact infant sleep (e.g. sleep displacement and possible effects of light on the circadian system and issues regarding the development of the melatonin system in infants).
  • Please expand on reasons why the effect of passive screen use (eg TV) compared to active screen use (smartphone) on sleep may differ and hence why the examination of the two media platforms?
  • Please include some comments on how the advent of PEDs has changed parenting practices.

Methods

  • No comment

Results:

  • To promote readability please organize the information in table 1 so that the differences between 1 and 3 year olds can be more easily compared (e.g. reorganize into three columns where column 1 = variable name(s), column 2 = age of 1 year data and column 3 = age of 3 years data). The Mother’s education can be included as text or a separate table.
  • The readability of the TV and DVD viewing time data reported in table 2 would also benefit by placing the independent variable sleep duration (age 1 vs age 3) and bedtime (age 1 vs age 3) in adjacent columns.
  • As the information is already included in the table no need to repeat the percentages (lines 133-138).
  • Are the values given in Figures 3-5 unadjusted or adjusted ORs? Reporting the adjusted ORs would be necessary.

Discussion:

  • Please note as a limitation that the evolution of and substantive uptake of web-enabled PED devices (e.g. smartphones etc) among children is relatively recent and therefore the data collected before 2016 may be under-reporting current trends (eg Škařupová K, Ólafsson K, Blinka L. The effect of smartphone use on trends in European adolescents’ excessive Internet use. Behaviour & Information Technology 2016; 35: 68-74. https://doi.org/10.1080/0144929X.2015.1114144 and Ghekiere A, Van Cauwenberg J, Vandendriessche A, Inchley J, Gaspar de Matos M, Borraccino A, Gobina I, Tynjälä J, Deforche B, De Clercq B. Trends in sleeping difficulties among European adolescents: Are these associated with physical inactivity and excessive screen time? International Journal of Public Health 2019; 64: 487-498. https://doi.org/10.1007/s00038-018-1188-1).
  • The study reports an association between screen time and shorter sleep/delayed phase at age 3. However, causality can’t be assumed from the present study. It is possible that children whose endogenous circadian rhythm is ‘naturally’ delayed (ie owls) may simply stay up later and therefore be more likely to have access to a PED device for longer periods in the evening. The issue of causality needs to be raised as a limitation.
  • It is noted that the effect of light exposure and especially PED screen light intensity on circadian phase in very young children is not known. This points to the need for safety guidelines regarding the use of PED devices in young children and as pointed out by several authors the need for public education campaigns.

Author Response

Dear Reviewer 1

Thank you for your important comment. We are grateful for the time and energy you have spent. Our response to the reviewer's comments is as follows:

Response to Reviewer 1

  • Introduction: Please expand the introduction and discuss the possible reasons why screen time may impact infant sleep (e.g. sleep displacement and possible effects of light on the circadian system and issues regarding the development of the melatonin system in infants).

We added three possible reasons why screen time may impact infant.

  1. Long-term use of TV/DVD, smartphone, and PED before going to bed delays bedtime.
  2. The stimulating program stimulates the brain and continues to secrete cortisol at night, making it difficult to fall asleep.
  3. The strong light of the screen at night suppresses the secretion of melatonin, and because of disturbing the circadian rhythm, late bedtime and short sleep duration.
  • Introduction: Please expand on reasons why the effect of passive screen use (eg TV) compared to active screen use (smartphone) on sleep may differ and hence why the examination of the two media platforms?

We added why TV/DVD and PED have different sleep effects. We also added the reasons

for investigating both TV/DVD and PED.

  1. PED is an attractive tool for kids because it gives them access to a variety of media at any time and allows them to touch and talk as well as see the screen.
  2. The PED is free to carry, so children can use it secretly for a long time with their parents.

For these reasons, We predicted that PED usage would result in short sleep duration and late bedtime than TV/DVD. Recently, PED usage in children has been increasing. We considered that PED may have a stronger effect on children's sleep than TV/DVD, and we sought to investigate whether there is a difference between TV/DVD and PED's sleep effect.

  • Introduction: Please include some comments on how the advent of PEDs has changed parenting practices.

    We have added the above content to the Introduction. We considered that the advent of PED could reduce opportunities for close communication between parents and children. The reason is that PED is an attractive toy for children, so children are always focused on PED and play alone, so parents are also very dependent on PED.

  • Results: To promote readability please organize the information in table 1 so that the differences between 1 and 3 year olds can be more easily compared (e.g. reorganize into three columns where column 1 = variable name(s), column 2 = age of 1 year data and column 3 = age of 3 years data). The Mother’s education can be included as text or a separate table.

    We have modified Table 1 as the reviewers pointed out. The Mother ’s education separated as Table 1-2.

  • Results: The readability of the TV and DVD viewing time data reported in table 2 would also benefit by placing the independent variable sleep duration (age 1 vs age 3) and bedtime (age 1 vs age 3) in adjacent columns.

We have modified Table 2 as the reviewers pointed out.

  • Results: As the information is already included in the table no need to repeat the percentages (lines 133-138).

It was pointed out that the content should be revised to make it easier to understand than Reviewer 2, so We revised it according to Reviewer 2.

  • Results: Are the values given in Figures 3-5 unadjusted or adjusted ORs? Reporting the adjusted ORs would be necessary.

The values shown in Figure 2-5 are adjusted ORs. We have revised the description in Figure 2-5.

  • Discussion: Please note as a limitation that the evolution of and substantive uptake of web-enabled PED devices (e.g. smartphones etc) among children is relatively recent and therefore the data collected before 2016 may be under-reporting current trends (eg Škařupová K, Ólafsson K, Blinka L. The effect of smartphone use on trends in European adolescents’ excessive Internet use. Behaviour & Information Technology 2016; 35: 68-74. https://doi.org/10.1080/0144929X.2015.1114144 and Ghekiere A, Van Cauwenberg J, Vandendriessche A, Inchley J, Gaspar de Matos M, Borraccino A, Gobina I, Tynjälä J, Deforche B, De Clercq B. Trends in sleeping difficulties among European adolescents: Are these associated with physical inactivity and excessive screen time? International Journal of Public Health 2019; 64: 487-498. https://doi.org/10.1007/s00038-018-1188-1)

As Reviewer 1 pointed out, we have added the above content as a limitation. We have also inserted citations.

  • Discussion: The study reports an association between screen time and shorter sleep/delayed phase at age 3. However, causality can’t be assumed from the present study. It is possible that children whose endogenous circadian rhythm is ‘naturally’ delayed (ie owls) may simply stay up later and therefore be more likely to have access to a PED device for longer periods in the evening. The issue of causality needs to be raised as a limitation.

As Reviewer 1 pointed out, we have added the above content as a limitation.

  • Discussion: It is noted that the effect of light exposure and especially PED screen light intensity on circadian phase in very young children is not known. This points to the need for safety guidelines regarding the use of PED devices in young children and as pointed out by several authors the need for public education campaigns.

As Reviewer 1 pointed out, We added the above content to the Introduction.

Reviewer 2 Report

Dear Authors,

Please see the following question from my point of view:

a) In the 'Introduction" section, I did see any explanation about the importance of such study or any reference from the previous works to suggest such study.

 b) Tables 1 and 2 show much information about the study but I did not find enough discussion to explain to us (readers).

c) I am sorry but the "Result' section needs to be rewritten to explain the results of this study to the readers to make it interesting. I can see some tables and figures, but I am sorry, the explanation is too short to understand the significance of each table and figure, how they are interesting as outcomes, how they are connected, how these observations support previous results from the previously published works? We need to keep in mind that neuroscience is a big branch and we all are not equally experts in this specific topic. So, as the authors, you need to explain your findings with logic to make us understand that the results are interesting.

d) Same for the "Conclusion" part as well. You need to justify why your outcomes are unique and interesting following the previously published works.

I will mark it as 'major revision'.  I am happy to revise the revised version.

thanks

Reviewer

Author Response

Dear Reviewer 2

Thank you for your important comment. We are grateful for the time and energy you have spent. Our response to the reviewer's comments is as follows:

Response to Reviewer 2

  • In the 'Introduction" section, I did see any explanation about the importance of such study or any reference from the previous works to suggest such study.

As Reviewer 2 pointed out, we have added to the Introduction section the importance of this study and the differences from previous studies.

  • Tables 1 and 2 show much information about the study but I did not find enough discussion to explain to us (readers).

We have revised the explanations in Tables 1 and 2 to make readers easier to understand, as Reviewer 2 pointed out.

  • I am sorry but the "Result' section needs to be rewritten to explain the results of this study to the readers to make it interesting. I can see some tables and figures, but I am sorry, the explanation is too short to understand the significance of each table and figure, how they are interesting as outcomes, how they are connected, how these observations support previous results from the previously published works? We need to keep in mind that neuroscience is a big branch and we all are not equally experts in this specific topic. So, as the authors, you need to explain your findings with logic to make us understand that the results are interesting.

As Reviewer 2 pointed out, we have rewritten the results in Figure 2-5 in detail to make it easier for the reader to understand. We have rewritten the results by classifying them into each item investigated so that the relationship between each analysis can be easily understood.

  • Same for the "Conclusion" part as well. You need to justify why your outcomes are unique and interesting following the previously published works.

As Reviewer 2 pointed out, we have rewritten the conclusions to make it easier to understand the unique points of this study and the interesting points following the previously published works.

Round 2

Reviewer 1 Report

There are only a small number of points that need consideration:

1) line 13 - the acronym PED is used for the first time in the Abstract without definition (see and swap with line 15)

2) line 35 - the acronym PED is used for the first time in the Introduction without definition (see and swap with line 52)

3) line 52 - UK data is given but more germane is there any Japanese data concerning PED and/or TV trends in young children (e.g material reported in lines 331-337). It would be best placed earlier.

4) line 52-69 - there is literature which can be cited to support these research questions and this paragraph would be strengthened by the inclusion of such references (e.g. Kostyrka‐Allchorne, K., Cooper, N. R., & Simpson, A. (2017). Touchscreen generation: children's current media use, parental supervision methods and attitudes towards contemporary media. Acta Paediatrica106(4), 654-662.)

5) line 77 - the Introduction contains a rationale for comparing between PED and TV but lacks a rational for examining the association between screen viewing time/ sleep habits at age 1 and screen viewing time/ sleep habits at age 3. There is a literature on this issue and a brief summary supporting the investigation would strengthen the paper (e.g. Kato, T., Yorifuji, T., Yamakawa, M., & Inoue, S. (2018). National data showed that delayed sleep in six‐year‐old children was associated with excessive use of electronic devices at 12 years. Acta Paediatrica107(8), 1439-1448. Atkin, A. J., Corder, K., & van Sluijs, E. M. (2013). Bedroom media, sedentary time and screen-time in children: a longitudinal analysis. International Journal of Behavioral Nutrition and Physical Activity10(1), 1-10.)

6) Tables 2.1 and 2.2  - I am aware of differences between disciplines but please see https://apastyle.apa.org/style-grammar-guidelines/tables-figures/tables regarding guidelines for the formatting the tables (especially the placement of grid lines)

7) line 295 - this study demonstrates an association not cause and effect, so I would advise replacing the word 'affected' with 'were associated'

Author Response

Dear Reviewer 1

Thank you for your important comment. We are grateful for the time and energy you have spent. We will resubmit the revised version of the manuscript. Our response to the reviewer's comments is as follows:

Response to Reviewer 1

  • line 13 - the acronym PED is used for the first time in the Abstract without definition (see and swap with line 15)

  As Reviewer 1 pointed out, we have revised it.

  • line 35 - the acronym PED is used for the first time in the Introduction without definition (see and swap with line 52)

As Reviewer 1 pointed out, we have revised it.

  • line 52 - UK data is given but more germane is there any Japanese data concerning PED and/or TV trends in young children (e.g material reported in lines 331-337). It would be best placed earlier.

As Reviewer 1 pointed out, we have revised it.

  • line 52-69 - there is literature which can be cited to support these research questions and this paragraph would be strengthened by the inclusion of such references (e.g. KostyrkaAllchorne, K., Cooper, N. R., & Simpson, A. (2017). Touchscreen generation: children's current media use, parental supervision methods and attitudes towards contemporary media. Acta Paediatrica106(4), 654-662.)

As Reviewer 1 pointed out, we have cited these references and further revised it.

  • line 77 - the Introduction contains a rationale for comparing between PED and TV but lacks a rational for examining the association between screen viewing time/ sleep habits at age 1 and screen viewing time/ sleep habits at age 3. There is a literature on this issue and a brief summary supporting the investigation would strengthen the paper (e.g. Kato, T., Yorifuji, T., Yamakawa, M., & Inoue, S. (2018). National data showed that delayed sleep in sixyearold children was associated with excessive use of electronic devices at 12 years. Acta Paediatrica107(8), 1439-1448. Atkin, A. J., Corder, K., & van Sluijs, E. M. (2013). Bedroom media, sedentary time and screen-time in children: a longitudinal analysis. International Journal of Behavioral Nutrition and Physical Activity10(1), 1-10.)

As Reviewer 1 pointed out, we cited these references and revised the content, including a rational for examining the association between screen viewing time/sleep habits at age 1 and screen viewing time/sleep habits at age 3.

  • Tables 2.1 and 2.2  - I am aware of differences between disciplines but please see https://apastyle.apa.org/style-grammar-guidelines/tables-figures/tables regarding guidelines for the formatting the tables (especially the placement of grid lines)

As Reviewer 1 pointed out, we added grid lines to Tables 1.1, 1.2 and 2.1, 2.2 and further changed the character placement.

7) line 295 - this study demonstrates an association not cause and effect, so I would advise replacing the word 'affected' with 'were associated'

As Reviewer 1 pointed out, we have revised it.

Reviewer 2 Report

Dear Authors,

This version is better than the previous version.

However, maybe some points were skipped from your sight to check:

a) in Intro section you need to justify why this study is important; were there any previous works where this type of study (even close idea) was discussed.

b) I can see the Result section has been improved a lot. Still, a good scientific explanation is important. Readers need to get the message of your works. For example, it took time for me to understand what are the roles of Table 2-1 and 2-2. You need to describe each table and figure and their significance.

c) In addition, you can also refer to some modeling work for future studies.

i) Modeling melanopsin-mediated effects of light on circadian phase, melatonin suppression, and subjective sleepiness

ii) A mathematical model of circadian rhythms and dopamine.

I will mark it as 'minor revision'.

Thanks

Reviewer

Author Response

Dear Reviewer 2

Thank you for your important comment. We are grateful for the time and energy you have spent. We will resubmit the revised version of the manuscript. Our response to the reviewer's comments is as follows:

Response to Reviewer 2

  1. in Intro section you need to justify why this study is important; were there any previous works where this type of study (even close idea) was discussed.

As Reviewer 2 pointed out, we have revised the introduction section to show the importance of this study, including its relevance to past studies.

  1. I can see the Result section has been improved a lot. Still, a good scientific explanation is important. Readers need to get the message of your works. For example, it took time for me to understand what are the roles of Table 2-1 and 2-2. You need to describe each table and figure and their significance.

As Reviewer 2 pointed out, we added titles above the results in Tables 2-1, 2-2 and revised the content to make it easier to understand the relevance and importance of each table and figure.

  1. c) In addition, you can also refer to some modeling work for future studies.
  2. i) Modeling melanopsin-mediated effects of light on circadian phase, melatonin suppression, and subjective sleepiness
  3. ii) A mathematical model of circadian rhythms and dopamine.

  As Reviewer 2 pointed out, we have cited these references and added to the discussion section that further study is needed in the future
